# Impact of COVID-19-Related Lockdown on Delivery and Perinatal Outcomes: A Retrospective Cohort Study

**DOI:** 10.3390/jcm11030756

**Published:** 2022-01-30

**Authors:** Thibaud Quibel, Norbert Winer, Laurence Bussières, Christophe Vayssière, Philippe Deruelle, Manon Defrance, Patrick Rozenberg, Jean Bouyer, Ninon Dupuis, Benoit Renaudin, Louise Dugave, Nathalie Banaszkiewicz, Charles Garabedian, Yves Ville

**Affiliations:** 1Department of Obstetrics and Gynecology, Poissy-Saint Germain Hospital, 78300 Poissy, France; manon.defrance.78@gmail.com (M.D.); patrick.rozenberg@ght-yvelinesnord.fr (P.R.); 2UVSQ, Inserm, Team U1018, Clinical Epidemiology, Centre de Recherche en Épidé-miologie et Santé des Populations (CESP), Paris Saclay University, 78180 Montigny-le-Bretonneux, France; jean.bouyer@inserm.fr; 3Obstetrics and Gynecology Department, Centre Hospitalier Universitaire de Nantes, 44035 Nantes, France; norbert.winer@chu-nantes.fr (N.W.); Nathalie.BANASZKIEWICZ@chu-nantes.fr (N.B.); 4UMR PhAN 1280 NUN INRAE F-44000 University Nantes, 44035 Nantes, France; 5Obstetrics, and Fetal Medicine and Surgery Department, Hôpital Necker-Enfants Maladies, AP-HP, 75007 Paris, France; laurence.bussieres@gmail.com (L.B.); benoit.renaudin@aphp.fr (B.R.); yves.ville@aphp.fr (Y.V.); 6EHU 7328 PACT, Université de Paris, 75006 Paris, France; 7Department of Obstetrics and Gynecology, Paule de Viguier Hospital, CHU Toulouse, 31059 Toulouse, France; christophe.vayssiere@gmail.com (C.V.); dupuis.ninon@gmail.com (N.D.); 8UMR1295 CERPOP (Centre for Epidemiology and Population Health Research), Team SPHERE (Study of Perinatal, Paedriatric and Adolescent Health: Epidemiological Research and Evaluation), Toulouse III University, 31062 Toulouse, France; 9Department of Obstetrics and Gynecology, University Hospital of Strasbourg, Avenue Moliere, 67000 Strasbourg, France; pderuelle@unistra.fr; 10CHU Lille, Department of Obstetrics, 59000 Lille, France; louise.dugave.etu@univ-lille.fr (L.D.); charles.garabedian@chru-lille.fr (C.G.); 11University Lille, ULR 2694 METRICS, 59000 Lille, France

**Keywords:** pregnancy, lockdown, pandemic COVID-19, perinatal and obstetrical issues

## Abstract

Objective: The magnitude and direction of effects on pregnancy outcomes of the lockdown imposed during COVID-19 have been uncertain and debated. Therefore, we aimed to quantify delivery and perinatal outcomes during the first nationwide lockdown due to the COVID-19 pandemic compared with the same durations of time for the pre- and post-lockdown periods. Study design: This was a retrospective cohort study of six university hospital maternity units distributed across France, each of which serves as the obstetric care referral unit within its respective perinatal network. Maternal and perinatal outcomes were compared between the lockdown period and same-duration (i.e., 55-day) periods before and after the 2020 lockdown (pre-lockdown: 22 January–16 March; lockdown: 17 March–10 May; post-lockdown: 11 May–4 July). We compared the overall rates of Caesarean delivery (CD), pre-labor CD, labor induction, operative vaginal delivery, severe postpartum hemorrhage (≥1 L), severe perineal tear, maternal transfusion, and neonatal mortality and morbidity (1- and 5-min Apgar scores < 7), hypoxia and anoxia (umbilical arterial pH < 7.20 or <7.10, respectively), and admission to a neonatal intensive care unit before discharge. Adjusted odds ratios were estimated using logistic regression, controlling for region of birth, maternal age category, multiparity, multiple pregnancies, diabetes, and hypertensive disorders. Results: The study sample consisted of 11,929 women who delivered consecutively at one of the six maternity units studied (4093 pre-lockdown, 3829 during lockdown, and 4007 post-lockdown) and their 12,179 neonates (4169 pre-lockdown, 3905 during lockdown, and 4105 post-lockdown). The maternal and obstetric characteristics of the women delivering during the lockdown period were alike those delivering pre- and post-lockdown on maternal age, parity, body mass index, rate of complication by hypertensive disorders or insulin-treated diabetes, and gestational age at delivery. Overall CD rates were similar during the three periods (23.6%, 24.8%, and 24.3% pre-lockdown, lockdown, and post-lockdown, respectively) and no outcome differed significantly during lockdown compared to pre- and post-lockdown. These findings were consistent across maternity units. Conclusion: The maternal and perinatal outcomes are reassuring regarding the performance of the health-care system during the COVID-19 lockdown studied. Such information is crucial, because additional COVID-19-related lockdowns might still be needed. They are also instructive regarding potential future pandemics.

## 1. Introduction

The 2019 novel coronavirus SARS-CoV-2 pandemic means that the world faced a serious and life-threatening infectious disease outbreak. In response to the World Health Organization declaration that COVID-19 was a public health emergency of international concern [1], many governments around the world adopted the most radical social distancing procedure, referred to as “lockdown”, to prevent the spread of the virus. In France, the first COVID-19 cases were diagnosed on 24 January 2020, and due to rapid disease progression, the initial lockdown was announced on 17 March 2020 and lasted until 10 May 2020 [2].

The lockdown period challenged the health-care system, notably at the maternity level, as it required many organizational changes to ensure maternal, fetal, and infant wellbeing. Indeed, referral maternity units within each region of France were selected to admit SARS-CoV-2-infected women. They were also asked to reorganize their logistical and human resources for this purpose. The lockdown may also have contributed to delays in seeking emergency care from fears of contracting SARS-CoV-2 at hospitals or not wanting to create added pressure on health-care workers during this difficult time. These factors may have increased the adverse consequences on maternal and perinatal health [3]. Management principles for COVID-19 in pregnancy were lacking, with no initial obstetrical recommendations announced until May 2020 [4]. There have also been concerns about intrapartum vertical transmission of SARS-CoV-2 infection, and a dearth of evidence about whether Caesarean delivery (CD) would reduce this.

For all these reasons, the first lockdown may have led to changes in pregnancy outcomes. Our aim was thus to quantify whether lockdown was associated with changes in delivery and/or perinatal outcomes.

## 2. Material and Methods

### 2.1. Setting

A nationwide lockdown in response to the COVID-19 pandemic was implemented in France on 17 March 2020 and continued until 10 May 2020. During this time, the population was required to stay at home or drastically limit their mobility, and to follow strict hygiene measures including handwashing and social distancing. This was accompanied by a shutdown of offices, shops, colleges, schools, and all institutions considered nonessential. After the government announced the end of the lockdown, nonessential services remained closed, and people were asked to continue restricting their social connections and limit mobility. This situation still applied at the end of our study period.

### 2.2. Study Design

We conducted a retrospective cohort study comparing three 55-day periods during 2020: (1) pre-lockdown (22 January–16 March), (2) lockdown (17 March–10 May), and (3) post-lockdown (11 May–4 July). We included all women with singleton or multiple pregnancies who gave birth at ≥24 weeks gestation with a birthweight ≥500 g (both criteria are needed for neonatal care at the maternity units studied) during the study period at one of six tertiary referral centers across France. The centers were in Lille in the north, Nantes in the west, Toulouse in the southwest, Strasbourg in the east, Poissy in the greater Paris area, and Necker in central Paris. These six centers represent 3.8% of all births in France (28,026 out of 740,000 in 2020).

During lockdown, the spread of COVID-19 through France was heterogeneous, and the pressure on the health-care system varied considerably. The situation was critical in eastern France and the Paris region, with intensive care units saturated with COVID-19 patients. In contrast, the north, west, and southwest regions were initially spared, to some extent [5].

### 2.3. Data Collection

A clinical observation checklist was first validated by research teams at the six maternity units to ensure data quality and homogeneity. Next, a local physician and/or research midwife at each maternity unit extracted the required data, including epidemiological, clinical, biological, maternal, and neonatal outcomes.

We collected the following data: maternal age (<35 years, 35–40 years or >40 years), parity (nullipara or multipara), singleton or multiple pregnancies, body mass index (underweight (<18 kg/m^2^), normal weight (18–25 kg/m^2^), overweight (>25–30 kg/m^2^), or obese (>30 kg/m^2^)), tobacco use (yes or no), gestational age at delivery (based on routine first-trimester ultrasound), diabetes requiring insulin therapy (gestational diabetes or preexisting diabetes), and hypertensive disorder (preeclampsia, superimposed preeclampsia, or preexisting). Neonatal variables were also collected: birthweight, 1- and 5-min Apgar scores, umbilical artery pH, admission to neonatal intensive care, and death. Data on stillbirths ≥24 weeks were also collected.

### 2.4. Outcomes

For each period, we compared gestational age at delivery, mode of labor onset (spontaneous, induction, or pre-labor CD), delivery mode (spontaneous vaginal, operative vaginal, or CD) and occurrences of vaginal perineal tear, severe postpartum hemorrhage (≥1 L), and maternal blood transfusion. Neonatal outcomes were also compared: 1- and 5-min Apgar score <7, umbilical artery pH <7.20 (hypoxia) and <7.10 (anoxia), admission to neonatal intensive care, neonatal mortality, and stillbirth.

We also compared indications for CD during the three periods: fetal distress, arrested labor, failure to progress, and other factors (e.g., breech/transverse position, history of CD, multiple pregnancies, and placenta previa).

### 2.5. Statistical Analysis

The results are shown as absolute values plus percentages for discrete variables and mean (standard deviation (SD)) or median (interquartile range (IQR)) for continuous variables. Separate comparisons were made between the outcomes during lockdown and those during the pre- and post-lockdown periods. Categorical and continuous variables were compared using χ^2^ tests and Student’s *t* tests, respectively. Adjusted odds ratios for overall CD, perineal tear, severe postpartum hemorrhage, and maternal transfusion, as well as for perinatal outcomes (neonatal intensive care admission, low 5-min Apgar score, neonatal anoxia, and stillbirth), were estimated by logistic regression, controlling for maternal age, multiparity, multiple pregnancies, diabetes, and hypertensive disorders. Because pressure on the health-care system varied by region, we performed a sensitivity analysis in which each outcome was assessed within each maternity.

We also analyzed the outcomes for high-risk pregnancies, which were complicated by hypertensive disorders or insulin-treated diabetes. To determine whether a delay in the management of these pregnancies had occurred, Kaplan-Meier survival curves were calculated to describe gestational age at delivery within each period.

All analyses were performed using R Studio version1.0.136. (https://www.rstudio.com/products/rstudio/download/, accessed on 25 November 2021). For all results, *p* < 0.05 was required for statistical significance.

### 2.6. Ethics

The Ethics Committee for Research in Obstetrics and Gynecology approved the study (OBS CEROG 2020-OBST-0705 on 12 October 2020). It was conducted in accordance with French legislation. Because standard care was provided at all tertiary centers and the dataset contained no information that could be used for patient identification, the study was exempt from informed consent requirements.

## 3. Results

From 22 January–4 July 2020, 12,154 women were consecutively registered at the six maternity units for delivering at gestational age ≥24 weeks; 174 women were excluded because they were managed as pregnancy terminations. The occurrence of home birth was similar during the three analysis periods. Our final sample was 11,929 women (4093 pre-lockdown, 3829 during lockdown, and 4007 post-lockdown) and 12,179 neonates (4169 pre-lockdown, 3905 during lockdown, and 4105 post-lockdown) (Figure 1).

Comparisons between lockdown and both pre- and post-lockdown periods are shown in Table 1. There were no significant differences regarding baseline characteristics or pregnancy complications, including hypertensive disorders and insulin-treated diabetes. Gestational age at delivery was also similar.

Maternal and perinatal outcomes are shown in Table 2 and Table 3. There were no statistically significant differences in adverse maternal or perinatal outcomes during the three periods. Labor onset mode (spontaneous, induced, or pre-labor CD), delivery mode (CD or operative vaginal), and postpartum hemorrhage rates also remained unchanged. Similarly, there were no differences in perinatal outcomes. When analyzed within each maternity unit separately, there were no significant differences across time periods (Appendix A).

Maternal and perinatal outcomes were unchanged when we analyzed the women attempting a vaginal delivery (Table 4). Indications for, and rates of, CD during labor also remained unchanged, suggesting that labor management did not change based on the lockdown phase.

Pregnancies complicated by hypertensive disorders or insulin-treated diabetes did not appear to cause any significant delay in adequate treatment, as gestational age at delivery was similar during the three periods studied (*p* = 0.70 and *p* = 0.50, respectively) and maternal and perinatal outcomes were similar over time among these population subsets (Figure 2 and Figure 3). The impact of lockdown on pre-term births and low birthweights have been the subject of a separate publication and are therefore not reported here [6].

## 4. Discussion

### 4.1. Main Findings

There was no apparent increase in adverse maternal or perinatal outcomes during the COVID-19-related lockdown period. These results were consistent across the six representative maternity units, which experienced unequal health-care system challenges from the SARS-CoV-2 outbreak. It also appears that high-risk pregnancies, such as those complicated by hypertensive disorders or diabetes, were managed as usual, as shown by the lack of shift in gestational age at delivery.

### 4.2. Clinical Interpretation

The impact of lockdown, which was imposed by many governments to limit the spread of the virus, has been studied in the contexts of several medical specialties. There was a real concern that lockdown would reduce access to the health-care system, delaying adequate care. However, from a population health perspective, its impact on emergency care seems to have been limited. Our findings herein suggest that maternal and perinatal outcomes were also unaffected by the lockdown. Regarding cardiovascular emergencies, Mesnier et al. showed a reduction in admission for acute myocardial infarction during lockdown in France, consistent with studies from the USA (California) and Italy [7,8]. The authors of said studies did not, however, find longer delays from symptom onset to admission or to invasive procedures, suggesting that cardiac sequelae were unaffected by lockdown. These findings are also consistent with studies evaluating the impact of lockdown on acute stroke outcomes. In Spain, admissions and thrombectomies performed for acute stroke were reduced by a quarter, without any indication that the quality of care had deteriorated [9]. These studies all suggest that during lockdown, there were population behavior modifications, but that these did not result in changes to the performance of the health-care system. Regarding obstetric care, the impact of lockdown is more difficult to analyze, as the issues may have been also influenced by a highly variable proportion of pregnant women infected by SARS-CoV-2, thus at increased risk of cesarean delivery, or admission in neonatal intensive care units for their newborns. However, modifications in the management of pregnancies or delay in maternal care during the lockdown may have also affected maternal and perinatal issues. Many studies have focused on the impact of the lockdown on pre-term births, low-weight births, and stillbirths [10,11,12]. In Iran, Ranjbar and al. found an increase of maternal admission in ICU, although the rates of complicated pregnancy with preeclampsia and diabetes remained unchanged and the proportion of infected women by COVID-19 was marginally low regarding the study population (6 pregnant women positive for COVID-19 out of 1216) [13]. Similarly, in India, Goyal et al. also suggested that a delay in seeking health care was one of the explanations of an increase rate in ICU admissions during the pandemic period [14]. Kc et al. recently described a decrease in institutional births during lockdown in Nepal, associated with significantly increased risks of pre-term birth, stillbirth, and neonatal mortality [15]. A significant reduction in intrapartum fetal heart rate surveillance, which may have occurred to reduce contact between health-care workers and pregnant patients, was presented as a possible explanation for these findings. Modifications in maternal and perinatal outcomes observed in low-income countries were also reported in high-income countries [16]. In Canada, Alshaik observed a decrease in the rates of very pre-term and very-low-weight-births during the lockdown, but no difference in spontaneous stillbirth. However, pregnant women who delivered during the lockdown period were diagnosed with gestational hypertension and chorioamnionitis more frequently. As induced pre-term births were more affected by the lockdown than spontaneous pre-term births, an increased rate in expectant management was suspected, which might also be part of the explanation of the increased rate of gestational hypertension and chorioamnionitis [16]. On the other hand, Kugelman et al. found no adverse maternal outcomes or neonatal morbidity, despite observed delays in arrival to the obstetrical emergency department and delivery room in Israel [17]. Finally, delivery practices also appear to have been modified during lockdown. Indeed, in Wuhan, although cesarean delivery rates remained unchanged, Li et al. observed that cesarean deliveries upon maternal request and for fetal distress were also more common during the lockdown [18]. In Australia, Rolnik et al. also observed a higher rate of cesarean delivery when pandemic restrictions were adopted, with a notable a trend towards more cesarean deliveries for the arrest of labor [19].

Herein, we found a decrease in deliveries during lockdown compared with pre- and post-lockdown that was consistent across the six maternity units. The home birth rate remained similar across periods; thus, we hypothesize that some women chose to deliver at small hospitals that did not manage patients with COVID-19, thinking they could reduce their risk of nosocomial contamination.

Interestingly, the CD rate observed in our study remained unchanged regardless of indication (notably including for non-reassuring fetal heart rate). Many decisions to proceed with CD are driven by the clinical needs of the mother, fetus, or both. However, health-professional-related factors are also involved [20], and lockdown was particularly stressful for these professionals. During the period studied, empirical evidence of maternal and fetal risks from COVID-19 and its management were still lacking, particularly for laboring women. Many caregivers were also concerned about their own risk of contamination. Within this context, stress could have had a significant effect on physicians’ medical decision-making. For example, they might have overcompensated and chosen a perceived safer delivery mode, such as CD, for borderline cases where, under normal circumstances, they would have proceeded with a spontaneous vaginal or instrumental delivery. We hypothesize that the health-care system, organizational design, and existing protocols and procedures among the six referral maternity units counterbalanced the negative effects of stress and allowed for the maintenance of the CD rate without changing perinatal outcomes, even in high-risk pregnancies. Therefore, we expect that organizational and system factors have a major role in controlling the impact of a lockdown on population health [21]. This point is especially important as, unfortunately, other pandemics will eventually occur. Thus, it is critical that we learn lessons from the COVID-19 pandemic now, in order to better prepare for those in the future.

Finally, and although we have described robust data herein showing that maternal and perinatal health remained stable during lockdown, this event may have affected maternal mental health. In Wuhan, Liu found that the COVID-19 outbreak and lockdown duration both aggravated prenatal anxiety [22]. Maternal stress hormones like glucocorticoids cross the placenta and can alter fetal programming via the hypothalamic–pituitary–adrenal axis, potentially leading to epigenetic changes through DNA methylation; thus, infants born during lockdown should be followed closely to ensure normal cognitive development [23,24].

### 4.3. Strengths and Limitations

The study strengths include its size and multicentric nature, allowing access to over 20,000 deliveries annually across a broad geographic area in France, and thus varying degrees of exposure to the first wave of the COVID-19 pandemic. However, all relevant outcomes remained consistent between maternity units, based on week-by-week assessments. Missing data occurred for less than 5% of all outcomes studied, allowing for a comprehensive overview of the impact of lockdown on pregnancies in a high-income country. Thus, the study provides comprehensive information about both the overall impacts of lockdown and those on high-risk pregnancies, which also appear to have been unaffected.

However, the study was not without limitations. First, the study setting limited the generalization of its results. All contributing maternity units were tertiary referral units, whose organization may differ from lower-level units. Second, we could not report the number of women with COVID-19 in our population because only acute, severe symptoms led to SARS-CoV-2 testing during this phase of the pandemic. Therefore, we could not assess the relations between asymptomatic or moderately symptomatic COVID-19 and obstetrical and perinatal outcomes. It remains unclear to date whether asymptomatic or moderately symptomatic women infected by SARS-CoV-2 are at risk for adverse pregnancy outcomes, including CD [25,26,27,28].

## 5. Conclusions

The findings reported herein are reassuring concerning the performance of the health-care system for maternal and perinatal outcomes during a national lockdown. This is crucial, as further lockdowns for COVID-19 may be necessary, and future pandemics are likely.

## Figures and Tables

**Figure 1 jcm-11-00756-f001:**
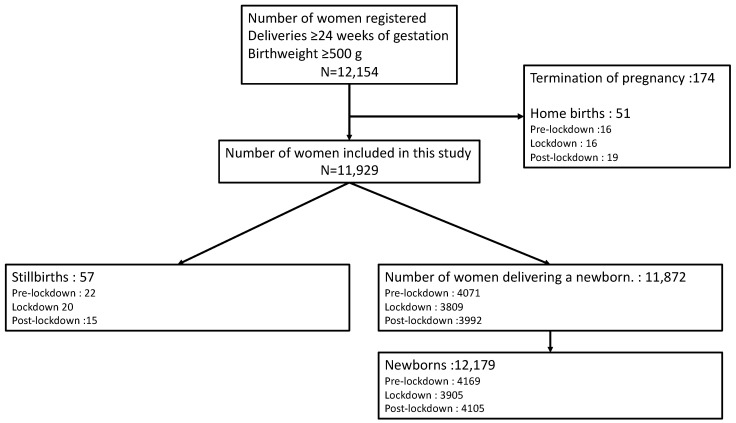
Flow chart for study cohort.

**Figure 2 jcm-11-00756-f002:**
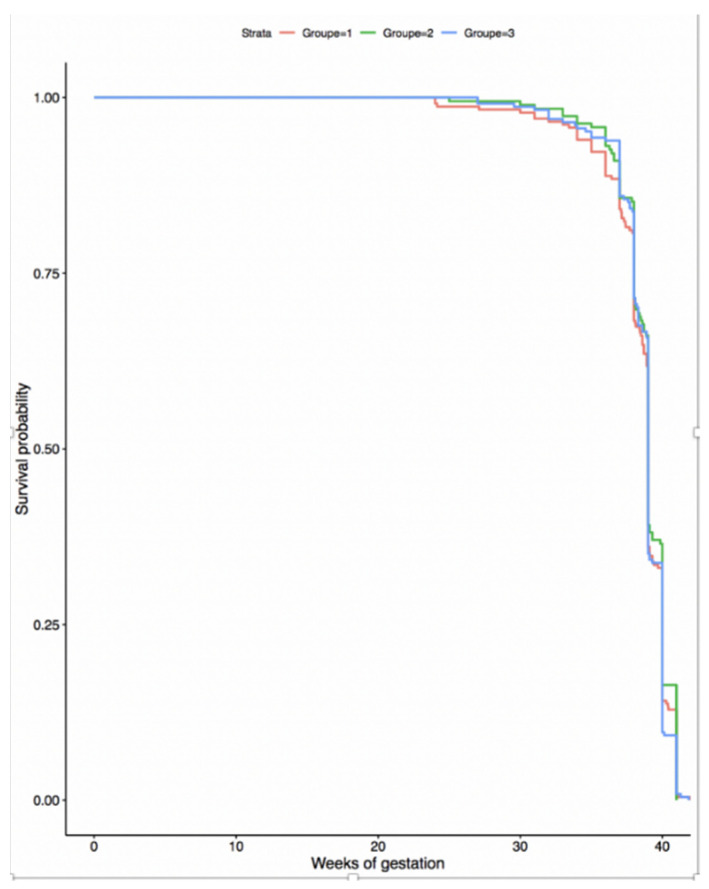
Gestational age at delivery for diabetic women treated with insulin by period studied. Red curve: pre-lockdown, green curve: lockdown period, blue curve: post-lockdown period.

**Figure 3 jcm-11-00756-f003:**
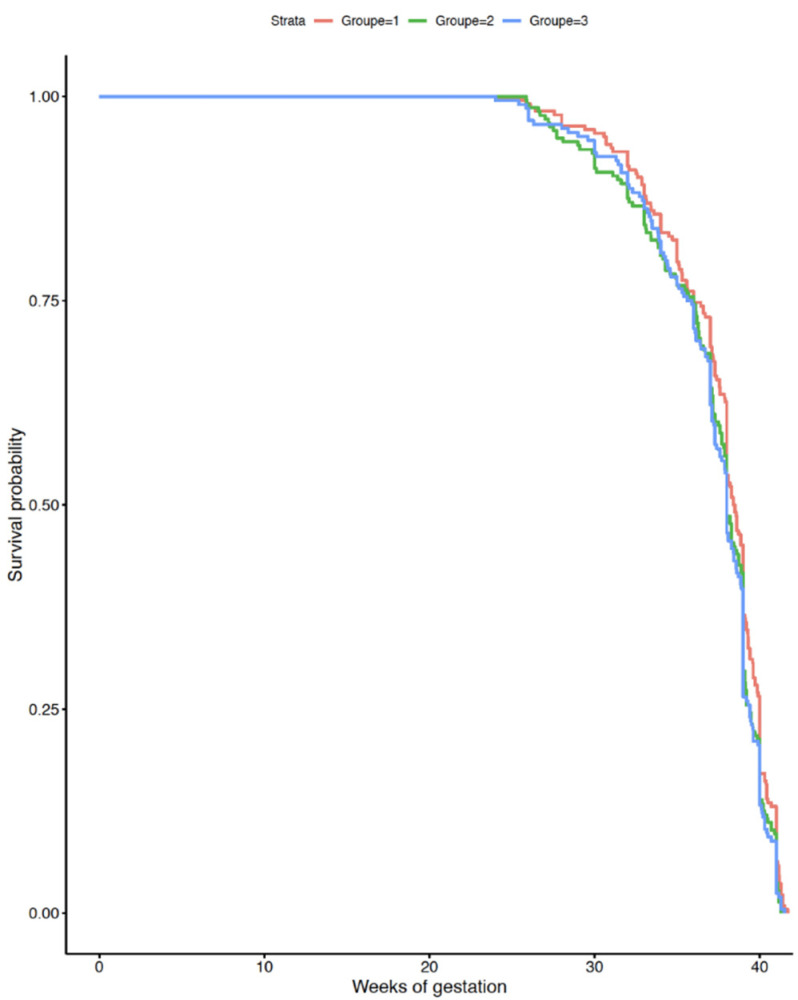
Gestational age at delivery for women with hypertensive disorders by period studied. Red curve: pre-lockdown, green curve: lockdown period, blue curve: post-lockdown period.

**Table 1 jcm-11-00756-t001:** Maternal characteristics and outcomes.

	Pre-Lockdown (*n* = 4093)	Lockdown (*n* = 3829)	Post-Lockdown (*n* = 4007)	P1	P2
Maternal age, median (Q1,Q3)	31.8 (28.0–35.4)	31.8 (28.1–35.5)	31.8 (28.1–35.5)	0.28	0.35
<35 years old, % (*n*)	72.9 (2985)	71.9 (2756)	71.5 (2867)		
35–40 years old, % (*n*)	20.2 (827)	20.7 (793)	21.4 (859)		
≥40 years old, % (*n*)	6.8 (281)	7.3 (280)	7.0 (281)		
Nullipara, % (*n*)	49.7 (1960)	50.8 (1877)	50.6 (1947)	0.31	0.83
Multiple pregnancies, % (*n*)	3.2 (133)	3.0 (115)	3.3 (132)	0.57	0.50
BMI (kg/m^2^)				0.82	0.42
<18.5	2.8 (103)	2.8 (99)	3.0 (108)		
18.5–25	59.8 (2227)	58.7 (2062)	60.5 (2199)		
26–30	22.9 (855)	23.7 (832)	22.4 (814)		
>30	14.4 (536)	14.7 (516)	14.1 (514)		
Diabetes with insulin, % (*n*)	8.9 (364)	7.7 (288)	8.8 (365)	0.17	0.09
Hypertensive disorders, % (*n*)	5.4 (220)	5.6 (216)	5.1 (204)	0.71	0.30
Active smoking, % (*n*)	11.3 (368)	10.2 (310)	10.1 (322)	0.23	0.91
Gestational age at delivery, WG, median (Q1,Q3)	39.3 (38.2–40.3)	39.3 (38–40)	39.3 (38.2–40.3)	0.66	0.27
<37^+0^ WG, % (*n*)	10.6 (435)	10.7 (435)	10.0 (402)	0.93	0.34
≥41 WG, % (*n*)	9.2 (377)	8.9 (340)	9.2 (367)	0.17	0.23

(Q1,Q3): Interquartile range. BMI: body mass index. WG: weeks of gestation. P1: comparison between lockdown and pre-lockdown. P2: comparison between lockdown and post-lockdown.

**Table 2 jcm-11-00756-t002:** Maternal and perinatal outcomes.

	Pre-Lockdown (*n* = 4169)	Lockdown (*n* = 3905)	Post-Lockdown (*n* = 4105)	P1	P2
Onset of labor				0.19	0.06
Spontaneous labor, % (*n*)	59.6 (2370)	57.9 (2138)	56.0 (2166)		
Labor induction, % (*n*)	28.0 (1111)	28.4 (1049)	30.1 (1166)		
Pre-labor CD, % (*n*)	12.4 (493)	13.6 (503)	13.8 (533)		
Overall CD, % (*n*)	23.6 (968)	24.8 (946)	24.3 (973)	0.28	0.68
Operative vaginal delivery *, % (*n*)	16.6 (519)	17.6 (506)	18.8 (571)	0.35	0.21
Perineal tear *, % (*n*)	1.1 (35)	1.0 (30)	1.3 (40)	0.86	0.37
Postpartum Hemorrhage > 1 L, % (*n*)	3.1 (125)	3.1 (119)	3.2 (130)	0.94	0.78
Transfusion, % (*n*)	0.9 (37)	1.1 (41)	0.9 (34)	0.52	0.51
Stillbirth, % (*n*)	0.5 (22)	0.5 (20)	0.4 (15)	0.81	0.41
1 min Apgar score < 7, % (*n*)	8.4 (337)	9.4 (351)	9.1 (356)	0.13	0.66
5 min Apgar score < 7, % (*n*)	1.8 (74)	2.4 (90)	2.3 (91)	0.09	0.86
Umbilical artery pH, % (*n*)				0.41	0.53
≥7.20	76.0 (3104)	76.2 (2,934)	77.3 (3148)		
7.10–7.19	19.4 (793)	18.8 (724)	17.9 (729)		
7.00–7.09	3.5 (144)	4.2 (160)	4.1 (168)		
<7.0	1.0 (41)	0.9 (33)	0.7 (27)		
NICU admission, % (*n*)	11.5 (478)	12.7 (495)	12.3 (503)	0.10	0.59
Neonatal death, % (*n*)	0.4 (16)	0.3 (11)	0.4 (18)	0.54	0.32

CD: caesarean delivery. NICU: neonatal intensive care unit. P1: comparison between lockdown and pre-lockdown. P2: comparison between lockdown and post-lockdown. * Ratio calculated among women with vaginal deliveries.

**Table 3 jcm-11-00756-t003:** Summary of odds ratios for maternal and perinatal outcomes.

	Pre-Lockdown vs. Lockdown	Post-Lockdown vs. Lockdown
CD rate *	1.06 [0.95–1.18]	1.02 [0.92–1.14]
Severe postpartum hemorrhage **	1.01 [0.76–1.28]	0.95 [0.74–1.23]
Perineal tear **	1.01 [0.66–1.56]	0.81 [0.53–1.21]
Maternal transfusion **	1.14 [0.73–1.80]	1.04 [0.65–1.64]
Apgar score 5 min < 7 **	1.33 [0.97–1.84]	0.93 [0.69–1.27]
pH < 7.10 **	1.10 [0.88–1.36]	1.04 [0.82–1.23]
NICU admission	1.15 [0.99–1.33]	0.96 [0.83–1.10]
Neonatal death	0.86 [0.37–1.82]	1.62 [0.77–3.56]
Stillbirth *	0.81 [0.39–1.37]	1.05 [0.52–2.12]

*: Adjusted for place of births, Maternal age (category), Multiparity, Multiple pregnancies, Diabetes, Hypertensive disorders, **: Adjusted for place of births, Maternal age (category), Multiparity, Multiple pregnancies, Diabetes, Hypertensive disorders and mode of delivery (caesarean yes/no).

**Table 4 jcm-11-00756-t004:** Outcomes among women undergoing a planned vaginal delivery.

	Pre-Lockdown (*n* = 3481)	Lockdown (*n* = 3187)	Post-Lockdown (*n* = 3332)	P1	P2
CD, % (*n*)	13.5 (467)	13.6 (432)	12.9 (429)	0.67	0.47
Operative vaginal delivery, % (*n*)	14.4 (499)	15.1 (480)	16.0 (533)	0.34	0.22
Perineal tear, % (*n*)	1.1 (38)	1.0 (31)	1.4 (46)	0.84	0.50
Postpartum hemorrhage, % (*n*)	2.6 (89)	2.7 (85)	3.0 (99)	0.77	0.17
Maternal transfusion, % (*n*)	0.9 (32)	1.0 (33)	0.9 (30)	0.73	0.66
5-min Apgar score < 7, % (*n*)	1.4 (48)	1.6 (51)	1.8 (60)	0.53	0.60
Arterial pH ≤ 7.10, % (*n*)	4.6 (157)	5.0 (158)	5.0 (167)	0.17	0.93
NICU admission, % (*n*)	8.1 (284)	9.0 (288)	9.2 (310)	0.23	0.77
Neonatal death, % (*n*)	0.3 (12)	0.2 (7)	0.3 (11)	0.46	0.54

CD: caesarean delivery. NICU: neonatal intensive care unit. P1: comparison between lockdown and pre-lockdown. P2: comparison between lockdown and post-lockdown.

## Data Availability

Data are available on request due to restrictions, e.g., privacy or ethical.

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
