# Peer review of "Impact of COVID-19-Related Lockdown on Delivery and Perinatal Outcomes: A Retrospective Cohort Study"

_jcm, 2022, doi:10.3390/jcm11030756_

Round 1
Reviewer 1 Report
Kugelman found no adverse maternal outcomes or neonatal morbidity, despite observed delays in arrival to the obstetrical emergency department and delivery room.(9) Ashish recently described a decrease in institutional births during lockdown in Nepal, and significantly increased risks of preterm birth, stillbirth, and neonatal mortality there.(10)
In one of study from India, there was an increased rate of cesarean section in COVID hospital. Also, the incidence of HIE and MSAF cases was more prevalent. Also found a possible vertical transmission in 9.4% of cases, which is higher than other reported literature. There is a rise in prematurity in SARS-CoV-2 positive pregnancy which might be due to an increased level of stress to the mothers. In this study, 60% of babies were delivered by a cesarean section, which is higher than the normal situation (https://doi.org/10.1016/j.pedneo.2021.05.004)
It seems the situation in developed country and in developing countries situation is different.
More references from India, China, Brazil should have been taken & compared to elucidate the true picture of the world vs the presentation from France.
Author Response
Dear Editor,
We thank the reviewer for his/her constructive comments, and we hope that this new manuscript will help to have a good overview of the maternal and neonatal health effects of the pandemic restrictions.
As requested by the reviewer, we provided more references studying this topic in the Discussion section, highlighting the differences in the impact of lockdown between countries. Modifications in management of pregnancies or delays in seeking for healthcare may have affected differently maternal and perinatal issues. These disparities appeared to be independent of the economic status (low-income or high-income) of the country. Here is the section we added to document the impact of the lockdown in different countries (page 9):
« Regarding obstetric care, the impact of lockdown is more difficult to analyze, as the issues may have been also influenced by a highly variable proportion of pregnant women infected by the SARS-CoV-2, thus at increased risk of cesarean delivery, or admission in neonatal intensive care units for their newborn. However, modifications in management of pregnancies or delay in maternal care during the lockdown may have also affected maternal and perinatal issues. Many studies were focused on the impact of the lockdown on preterm births, low-weight births and stillbirths.(10-12) In Iran, Ranjbar and al. found an increase of maternal admission in ICU, although the rates of complicated pregnancy with preeclampsia and diabetes remained unchanged and the proportion of infected women by Covid-19 was marginally low regarding the study population (6 pregnant women positive for Covid-19 out of 1216).(13) Similarly, in India, Goyal and al. also suggested that a delay in seeking health care was one of the explanations of an increase rate in ICU admissions during the pandemic period.(14) Kc et al. recently described a decrease in institutional births during lockdown in Nepal, associated with significantly increased risks of preterm birth, stillbirth, and neonatal mortality.(15) A significant reduction in intrapartum fetal heart rate surveillance, which may have occurred to reduce contact between health-care workers and pregnant patients, was presented as a possible explanation for these findings. Modifications in maternal and perinatal outcomes observed in low-income countries were also reported in high-income countries.(16) In Canada, Alshaik observed a decrease in rates of very preterm and very-low -weight-births during the lockdown, but no difference in spontaneous stillbirth. However, pregnant women who delivered during the lockdown period were diagnosed with gestational hypertension and chorioamnionitis more frequently. As induced preterm births were more affected by the lockdown than spontaneous preterm births, an increased rate in expectant management was suspected which might also be part of the explanation of the increased rate of gestational hypertension and chorioamnionitis.(16) On the other hand, Kugelman et al. found no adverse maternal outcomes or neonatal morbidity, despite observed delays in arrival to the obstetrical emergency department and delivery room in Israel.(17) Finally, delivery practices also appear to have been modified during lockdown. Indeed, in Wuhan, although cesarean delivery rates remained unchanged, Li and al. observed that cesarean deliveries on maternal request and for fetal distress were also more common during the lockdown.(18) In Australia, Rolnik et al. also observed a higher rate of cesarean delivery when pandemic restriction were adopted, with notably a trend towards more cesarean deliveries for arrest of labor.(19)”
For clarity, we also specified in the Results section on page 6 that:
“The impact of lockdown on preterm births and low birthweights have been the subject of a separate publication and are therefore not reported here (6).”
Garabedian C, Dupuis N, Vayssière C, Bussières L, Ville Y, Renaudin B, Dugave L, Winer N, Banaszkiewicz N, Rozenberg P, Defrance M, Legris ML, Quibel T, Deruelle P. Impact of COVID-19 Lockdown on Preterm Births, Low Birthweights and Stillbirths: A Retrospective Cohort Study. J Clin Med. 2021 Nov 30;10 (23):5649.

Reviewer 2 Report
Although this paper is a multi-center study with a significant number of participants, it does not provide any interesting, nor novel findings. The title is misinformed, as this manuscript does not tell us the impact of COVID-19 on the perinatal and delivery outcomes in women infected with the disease. It is a retrospective snapshot of descriptive data comparing numbers during the pre/lockdown/post times which does not provide any useful findings or conclusions.
Author Response
Dear Editor
We would like to thank the reviewer for his/her comments. Here are the answers we would like to share with him/her.
The reviewer suggested that the title is inappropriate with the subject of this article. However, we disagree with his point of view, as the title reflects the objective studied in this manuscript, which was to study the impact of a lockdown on maternal and perinatal issues. Specifically, our aim was not to study the impact of the Covid-19 disease on pregnant women and their newborns, as suggested by the reviewer, but to analyze whether maternal and perinatal health were affected when the lockdown was pronounced.
The impact of pandemic restrictions on maternal issues and overall perinatal issues is a matter of great importance, since further lockdowns for COVID-19 may be necessary, and future other pandemics are likely. Most studies were focused on the impact of the lockdown on preterm births, low weights births and stillbirths, showing discrepancies between countries. Similarly, maternal and perinatal outcomes are also conflicting, notably since delays for seeking maternal health care or modifications in obstetric and delivery practices may have been different from one country to another. Therefore, as it was also requested by the other reviewer, we provided in the revised manuscript more references studying the impact of the lockdown on these outcomes, which we hope will be helpful to illustrate the disparities between countries which appeared to be independent of their economic status (low-income or high-income).

Round 2
Reviewer 2 Report
I acknowledge the authors feedback regarding the reviewer comments. Many thanks. Overall the paper is improved following amendments.